# Surface Modification via Dielectric Barrier Discharge Atmospheric Cold Plasma (DBD–ACP): Improved Functional Properties of Soy Protein Film

**DOI:** 10.3390/foods11091196

**Published:** 2022-04-20

**Authors:** Zhibing Li, Shanggui Deng, Jing Chen

**Affiliations:** 1Zhejiang Provincial Key Laboratory of Health Risk Factors for Seafood, Collaborative Innovation Center of Seafood Deep Processing, College of Food and Pharmacy, Zhejiang Ocean University, Zhoushan 316022, China; omar18355099192@163.com (Z.L.); dengshanggui@163.com (S.D.); 2Key Laboratory of Health Risk Factors for Seafood of Zhejiang Province, Zhoushan 316022, China

**Keywords:** atmospheric cold plasma, soy protein, film, modification, properties

## Abstract

Atmospheric cold plasma (ACP), a novel technology, has been widely adopted as an efficient approach in surface modification of the film. The effect of ACP treatment on the physicochemical and structural properties of soy protein film were investigated. As a result, the optimal conditions for the preparation of the film were determined for soy protein (10%), glycerol (2.8%), ACP treatment at 30 kV for 3 min, on the basis of elongation at the break, and water vapor permeability. Under the optimal conditions, the ACP–treated films exhibited enhanced polarity according to the increased values of solubility, swelling index, and moisture content, compared with the untreated counterpart. An increase in the hydrophilicity is also confirmed by the water contact angle analysis, which decreased from 87.9° to 77.2° after ACP pretreatment. Thermostability was also improved by ACP exposure in terms of DSC analysis. SEM images confirmed the tiny pores and cracks on the surface of film could be lessened by ACP pretreatment. Variations in the Fourier transform infrared spectroscopy indicated that some hydrophilic groups were formed by ACP pretreatment. Atomic force microscopy data revealed that the roughness of soy protein film which was pretreated by ACP was lower than that of the control group, with an Rmax value of 88.4 nm and 162.7 nm for the ACP- treated and untreated samples, respectively. The soy protein film was characterized structurally by FT–IR and DSC, and morphological characterization was done by SEM and AFM. The soy protein film modified by ACP was more stable than the control group. Hence, the great potential in improving the properties of the film enables ACP treatment to be a feasible and promising alternative to other modification methods.

## 1. Introduction

With the increasing harm of plastics to the environment, the security of the living environment has gained considerable attention [1]. Nowadays, searching for alternatives to plastics has become a hot issue. Bio-based materials evoked intensive interests due to ease of processability, abundant resources, and eco-friendliness. Among them, soy protein, a widely distributed vegetable protein, has been undoubtedly considered as the most important material, ascribed to its extremely low price, easy availability, high protein content, and excellent film-forming properties [2]. Hence, films made from soy protein can be applied in edible packaging and food preservation. It has gradually attracted attention as an alternative to plastics [3]. Nevertheless, films prepared from soy protein are difficult to be employed in food packaging unless its drawbacks, such as high rigidity, easiness to grow bacteria, and poor stability, are improved [4,5].

In this regard, a series of strategies has been developed to modify the soy protein film, thus to achieve desirable functions. Among them, addition of exogenous compounds has been extensively utilized to improve the film properties. Glycerol, 1- and 2- propylene glycol, and sorbitol were reported to be added into the soy protein solution to enhance the mechanical strength of the film [6]. The supplement of citric acid, nalidixic acid [7], and mandelic acid [8] to soy protein could enhance the antibacterial properties of the film. The barrier properties of the soy protein film were improved through adding CaCl_2_ [9], ZnSe [10], and ZnS [11] nanoparticles [12]. However, chemical additives may have adverse health effects, such as toxicity, food allergies, and impaired nutrition. Heating, as the most common physical modification of protein, would lead to the dissociation of quaternary structures and protein subunit denaturation [13]. Other solvent-free techniques, such as pressure treatment, ultrasound, pulsed electric field, microwaves, and gramma irradiation, have also been reported to modify protein functionally by destroying the internal chemical bonds of protein and making it re-cross-linking [14]. These non-thermal processes demonstrated great potential in protein modification, but some of them still cannot meet the requirement of desirable functions. Therefore, it is necessary to develop a novel approach for current food and material industries.

Atmospheric cold plasma (ACP) is a novel non-thermal and pollution-free technology [15]. It can produce a large number of reactive particles, including charged particles, free radicals, photons, and various radiations, through high-voltage discharge under normal temperature conditions [16]. ACP has been widely applied in the food industry owing to its high microbial decontamination effectiveness. As reported, ACP exhibited excellent eliminating effects on *Escherichia coli* [17], *Listeria* [18], *Salmonella*, *Shigella* [19], *Staphylococcus aureus* [20], and *Thermobacteria* [21]. Besides, it is worth noting that cold plasma is capable of selective modification of starch and protein [16]. As a promising modification alternative, ACP displayed several advantages, including no generation of thermal damage, free of additives, and inexpensive operation costs [22]. Moreover, ACP treatment was also proved to be an efficient modification technology without destroying bulk attributes, since the treatment only affected a few nanometers below the surface of the materials [23]. ACP treatment for wheat flour at 80 kV for 5–30 min resulted in an improvement of hydration properties [24]. It was demonstrated that emulsion stability and water holding capacity were prominently enhanced for peanut isolated protein subjected to ACP treatment [25]. Pankaj et al. [26] observed that the roughness of polylactic acid film was increased after ACP treatment. It was also observed that strength and surface hydrophobicity of zein film could be reinforced by ACP treatment [27]. Similar results were also obtained by Oh et al. [28], who described that ACP treatment resulted in improved tensile strength and a moisture barrier of soybean meal-based edible film.

Although several studies have reported that cold plasma was able to interact with protein film, the effect of cold plasma technology on the soy protein film has been scarcely investigated [29]. In order to provide a better understanding of ACP interactions with this edible film, in this study, dielectric barrier discharge (DBD) employing atmospheric air as a working gas was used to generate CP. Soy protein was subjected to ACP treatment prior to preparing the film. The physical properties, thermal properties, and microscopic morphology of the soy protein film were evaluated.

## 2. Materials and Methods

### 2.1. Materials

Soy protein was purchased from Linyi Shansong Biological Products Co., Ltd. (Linyi, China). All chemicals were of analytical grade purity and commercially available.

### 2.2. ACP Treatment of Soy Protein Solution

The protein solution was produced in four steps. First, 100 mL of deionized water was mixed with 10 g of soy protein powder. The pH of the soy protein solution was adjusted to 9.5–10, which was placed in a 40 °C water bath for 15 min and then stirred with a magnetic stirrer for 15 min. The protein solution, after cooling for 30 min, was filtered with three layers of gauze to remove air bubbles. The protein solution was subjected to DBD–ACP (Phenix Technologies, Accident, MD, USA) treatment which was reported in our previous study [30]. The schematic diagram of the device is shown in Figure 1.

Parameters including ACP treatment voltage and exposure time were investigated. Soy protein was treated at a fixed exposure time (3 min) with varying voltages of 10, 20, 30, 40, and 50 kV. The exposure time (1, 2, 3, 4, 5 min) was also assessed at a fixed voltage (30 kV).

### 2.3. Preparation of Soy Protein Film

Soy protein film was prepared by hot-air drying based on the method provided by Kumar et al. [8], with slight modification. Glycerol was added to the soy protein solution after ACP treatment. The sample was stirred with a magnetic stirrer for 15 min and then it was allowed to stand for 1 h. Three layers of gauze were used to remove scum in the soy protein solution. About 40 mL of the treated sample solution was collected on a constant-temperature platform and poured into a 20 cm × 20 cm silicone mold. The samples were heated at 35 °C on a constant temperature platform for 4 h and then transferred to a desiccator. Hot air was blown until the solvent was completely volatilized (25 °C). The soy protein film was stored in a desiccator at 25 ± 0.1 °C (relative humidity 50%) for 24 h prior to the test. Film prepared from soy protein without ACP treatment was considered as the control group.

The amount of soy protein (8%, 9%, 10%, 11%, and 12%) was investigated with a fixed concentration of glycerol (2.4%). Meanwhile, the amount of glycerol (1.6%, 2.0%, 2.4%, 2.8%, and 3.2%) was also evaluated with a fixed amount of soy protein (10%).

### 2.4. Mechanical Properties of Soy Protein Film

Mechanical properties of the soy protein film were tested by an electric tensile testing machine (Dongguan Zhitake Precision Instrument Co., Ltd., Dongguan, China), according to Dong et al. [27], with a slight modification. A special knife was used to cut the film sample with a width of 24 mm and length of 50 mm prior to measurement. The test speed was 10 mm/min. Elongation at the break (E, %) was determined using Equation (1).
(1)$$E=\frac{L_{1}-L_{0}}{L_{0}}\times 100\%$$
where L_0_ (mm) is the initial length of the film, and L_1_ (mm) is the length at break.

### 2.5. Water Vapor Permeability (WVP)

The moisture permeability of the control and ACP -treated soy protein film was evaluated by water vapor permeation instrument (PERMATRAN -W 1/50, Mocon, Minneapolis, MN, USA) according to the procedure of Wiles et al. [31], with minor modifications. The film (100 mm^2^ film area) was placed uniformly in a closed container with a relative humidity of 50% ± 1% at 25 °C and for 48 h. Water vapor permeability (WVP) was expressed as Equation (2).
(2)$$\mathrm{WVP}=\frac{\mathrm{WVTR}\times t}{P_{w}}$$
where WVP is the water vapor transmission coefficient (g·mm/m^2^·d·kPa), WVTR is the water vapor transmission rate (g/m^2^·d), t is the average thickness of the film (mm), and P_W_ is the water pressure difference of vapor passing from one side of the film to the other side (kPa).

### 2.6. Moisture Content (MC)

Soy protein films were weighed before and after drying in an electric oven at 110 °C until a constant weight was obtained [32]. MC was expressed as Equation (3).
(3)$$\mathrm{MC}\left(\%\right)=\frac{M-M_{1}}{M}\times 100\%$$
where MC is the moisture content (%), M is the initial weight of soy protein film (g), and M1 is the constant weight of soy protein film.

### 2.7. Solubility (So)

The solubility in water of soy protein film was determined according to the method described by Peng et al. [32]. Soy protein film (2 cm × 2 cm) was soaked in 30 mL of pure water. Then, the sample was placed in an oscillation box at 25 °C for 24 h and dried to a constant weight. The solubility was calculated as Equation (4):(4)$$\mathrm{So}\left(\%\right)=\frac{M-M_{1}}{M}\times 100\%$$
where So is the solubility (%), M is the initial weight of soy protein film (g), and M_1_ is the constant weight of soy protein film (g).

### 2.8. Swelling Index (Si)

The swelling index (Si) reflects the water absorption capacity of soy protein film [33]. Film pieces (2 cm × 2 cm) were dried at 70 °C for 24 h in a vacuum oven to obtain the initial dry weight. The sample was transferred into a 100 mL beaker with 30 mL pure water and soaked in the water at 25 °C for 24 h. Subsequently, the moisture on the film surface was sucked with filter paper and weighed up. The swelling index is expressed as Equation (5).
(5)$$\mathrm{Si}\left(\%\right)=\frac{M_{1}-M}{M}\times 100\%$$
where Si is the swelling index (%), M_1_ is the quality of soy protein film after soaking (g), and M is initial dry weight of soy protein film (g).

### 2.9. Differential Scanning Calorimetry (DSC)

The thermodynamic properties of soy protein film were determined by NETZSCH 200 F3 DSC (NETZSCH, Selb, Germany), adopting the protocol of Dong et al. [27]. Approximately 8–12 mg of the sample was weighed and placed in an aluminum pan with an empty aluminum pan as reference. Samples were subjected to two heating–cooling cycles from 20 °C to 200 °C at a rate of 10 °C/min. Denaturation temperature (Td), glass transition temperature (Tg), heat, and Denaturation Enthalpy were calculated with 89TA-60WS software.

### 2.10. Scanning Electron Microscopy (SEM)

The surface morphology of the control and ACP-treated soy protein film were recorded by a cold-field emission scanning electron microscope (Su8010, Hitachi, Japan) [3]. Prior to the measurement, all samples were fixed on the stage and coated with a layer of gold for 5 min. Afterwards, the sample was placed on the scanning electron microscope under an acceleration voltage of 5.0 kV to capture images.

### 2.11. Fourier Transform Infrared Spectroscopy (FTIR)

The infrared spectrum of the soy protein film was analyzed by a Fourier spectrometer (FT/IR-650, Thermo Nicolet Inc., Waltham, MA, USA) within a range of 4000 to 500 cm^−1^, at a resolution of 4 cm^−1^ [5]. The KBr tablet method was adopted, and a mixture composed of soy protein film (1 mg) and KBr (50–100 mg) was grounded into fine powder. The powder was compressed into tablets and placed in the sample chamber.

### 2.12. Atomic Force Microscopy (AFM)

Surface morphology of the soy protein film was assessed following the procedure adapted from Pankaj et al. [26], with some modifications by Atomic Force Microscope (Dimension Icon, Bruker, Germany) operating in intermittent contact (tap) mode. Images were collected at a fixed scan rate of 0.5 Hz. The AFM images provided topographic images as well as quantitative data. The scanning area was 25 μm^2^.

### 2.13. Water Contact Angle (WCA)

Water contact angle was used to evaluate the hydrophilicity and hydrophobicity of the surface of the soy protein film [8]. The measurement was conducted by Optical Contact Angle Measuring Device (OCA20, Germany Dataphysics, Filderstadt, Germany). About 5.5 μL of distilled water was dropped on the film surface with a micro-syringe.

### 2.14. Statistical Analysis

All experiments were carried out in triplicate. Analysis of variance (ANOVA) was used to evaluate differences among means, and statistical software was adopted to compare the means of different groups by Turkey’s test at the 5% significance level (*p* < 0.05). Data for each treatment were collected in triplicate, and their results were expressed as average ± standard deviation and analyzed using the descriptive statistics function in Origin 8.5 software (Origin Lab, Northampton, MA, USA).

## 3. Results

### 3.1. Factors Affecting Elongation at Break and Water Vapor Permeability of the Soy Protein Film

The effect of several parameters, including soybean protein concentration, glycerol concentration, ACP treatment voltage, and exposure time, were investigated in terms of elongation at break and water vapor permeability, which are regarded as important indexes to display the packaging performance of soy protein film. As is known, elongation at the break was used to evaluate the mechanical properties of the film, while water vapor permeability (WVP) can be adopted to indicate the barrier property.

#### 3.1.1. Effect of Soybean Protein Concentration

Films with different amounts of soybean protein were prepared and the optimal concentration was investigated when glycerol concentration was fixed as 2.4%, and ACP treatment voltage and time were set as 30 kV and 3 min, respectively. As depicted in Figure 2, elongation at break and water vapor transmittance of the soy protein film were obtained under different soybean protein concentrations. As the soy protein concentration increased, the elongation at the break enhanced at first and then declined for the ACP-treated film, while a steady decrease was observed for the counterpart. The maximum value of 227.37% was achieved for the ACP-treated film when the content of soybean protein was 10%. As reported, soy protein concentration was highly related to the properties of the protein film [30]. The poor toughness was a serious problem, which limited the widespread utilization of soy protein film [34]. For all the tested concentrations, ranging from 8% to 12%, elongation at the break of the treated film was significantly higher (*p* < 0.05) than that of the control group, demonstrating that cold plasma treatment was efficient in modifying the target protein. Elongation at the break of the treated group was 143.73% higher than that of the control group. Meanwhile, according to Figure 2b, water vapor permeability of the film exerted an extremely slight decrease with the increase in soy protein concentration both for the ACP-treated and untreated group. However, an obvious enhancement of WVP was detected between the ACP-treated and untreated group for all the tested soy protein concentrations, confirming the modification efficiency by ACP. Hence, the optimum soy protein content was selected as 10%.

#### 3.1.2. Effect of Glycerol Concentration

As a plasticizer, glycerol is commonly used in edible films due to its small molecular size and abundant functional hydroxyl groups. Hence, the addition of glycerol is of great importance in the formation of soy protein film, and the optimal concentration was evaluated under a fixed soy protein content of 10%, and ACP treatment conditions of 30 kV for 3 min. As shown in Figure 3a, with the increase of glycerol concentration, the elongation at the break increased first and then decreased for the ACP–treated film, while a gradual increase was detected for the untreated film. The elongation at the break reached the peak (223.32%), with the glycerol addition of 2.8%. It was found that when the addition of glycerol was 1.6%, the film was difficult to uncover and easily broken. Furthermore, the rigidity and brittleness of the soy protein film turned out to be relatively high. The packaging performance of the film became extremely poor. As is reported, glycerol was embedded in the soy protein to enhance the plasticity of the film, which made the film soft and elastic [35]. Moreover, as the glycerol concentration increased, WVP of soy protein film displayed a gradual downward trend both for the ACP–treated and untreated samples (Figure 3b). It was observed that WVP dropped from 13.48 g·mm/m^2^·d·kPa to 6.02 g·mm/m^2^·d·kPa when the glycerol concentration raised from 1.6% to 3.6%, which was predominantly due to the higher capacity of higher concentration of glycerol in promoting protein cross-linking. Cracks and numerous pores would appear after the protein was prepared in the film, especially for the films with low concentrations or without the addition of glycerol. The compactness of the film would be improved owing to the supplement of glycerol, which was reported to be embedded in the gaps of the soy protein [5]. No significant difference (*p* > 0.05) of WVP was recorded between the treated and control samples when the glycerol content varied from 2.4% to 3.2%. Thus, taken with the effect on elongation at the break, the optimal glycerol content of soy protein film was finally determined to be 2.8%. Similarly, ACP treatment resulted in a remarkable increase in elongation at the break and a notable decrease of WVP, compared with the counterpart, further implying that cold plasma is capable of modifying the protein film.

### 3.2. Effects of Different ACP Treatment Voltages on Soy Protein Film

The effect of ACP voltage level on the mechanical properties of the soy protein film was investigated at a fixed treatment time of 3 min with the optimal soybean and glycerol concentration of 10% and 2.8%, respectively. As illustrated in Figure 4a, elongation at the break increased at a rapid rate and subsequently decreased at a fast rate, with a maximum value of 211.53% at 30 kV. An increase in the treatment voltage exhibited a profitable effect, which could improve the elongation at the break of soy protein film and help the formation of gels. In this study, ACP could promote the interaction between proteins, which was one of rationales for the capacity of modifying soy protein film [25]. Higher treatment voltage, however, would decrease the elongation at the break of the soy protein film. The possible reason is that ACP treatment would destroy the active sites on the surface of soy protein at higher voltage levels, resulting in the denaturation of some soy protein and disability of participation in the cross-linking reaction. Meanwhile, WVP of the treated soy protein film decreased slightly at first but increased significantly (*p* < 0.05) after the treatment voltage exceeded 30 kV (Figure 4b). WVP was found to be 7.07 g·mm/m^2^·d·kPa when the soy protein was treated at 30 kV, while this value boosted to 40 kV, and WVP was increased to 9.29 g·mm/m^2^·d·kPa. The deduction made was that active particle (O^2−^, O^2+^, H_3_O^+^) interacted with the surface of soy protein, leading to an increase in WVP [29]. This phenomenon was consistent with that of Pankaj et al. [26], who also found an increase in the water vapor transmission rate of the film made from polylactic acid under a higher voltage level. Based on the comprehensive appeal, it is concluded that 30 kV is the best ACP processing voltage.

### 3.3. Effects of Different ACP Treatment Times on Soy Protein Film

ACP treatment time was also optimized by varying from 1 min to 5 min at the above obtained optimal conditions. Variations in the elongation at the break and water vapor transmittance of soy protein films treated with different voltages were displayed in Figure 5. The elongation at the break increased significantly (*p* < 0.05) as the treatment time was prolonged from 0 to 3 min. After 3 min of exposure to ACP at 30 kV, the elongation at the break reached the peak value (199.64%). This finding was in agreement with the results of Jahromi et al. [36], who also observed an increase of casein film in elongation at the break after ACP treatment. The elongation at the break no longer increased when the treatment time was further extended to 5 min. In contrast, it reduced significantly (*p* < 0.05) at the extension of exposure time. The structure of protein was destroyed and the molecules were broken down, which in turn caused the proteins to be unable to cross-link well, with extended treatment time. In the case of WVP, it slightly declined and then significantly enhanced (*p* < 0.05) with the increased treatment time (Figure 5b). Extended exposure to ACP caused modification of the soy protein to a large extent, which was harmful to the interaction between proteins. A similar phenomenon was also reported by Dong et al. [27], who found that the original smooth surface of zein gradually disappeared, and a considerable number of irregular distortions and ruptures occurred after ACP treatment. To sum up, the exposure time of 3 min was selected for further analysis.

Based on the above observations, it was concluded that the optimum conditions for preparation of the film were as follows: soy protein concentration of 10%, glycerol concentration of 2.8%, treatment voltage of 30 kV, and treatment time of 3 min. The characteristics of both the control and ACP–treated film were investigated by DSC, FTIR, SEM, AFM, and WCA.

### 3.4. Solubility, Swelling Index, and Moisture Content of Soy Protein Film

Moisture content, solubility, and swelling index are three important characteristics of protein film, which can positively reflect the hydrophilicity of soy protein film. The results of solubility, swelling index, and water content of soy protein film were listed in Table 1. The control film displayed a lower solubility and swelling degree than those of the ACP-treated film. As reported, the solubility mostly depended on the hydrophilic properties of films, while the swelling index was relevant to water diffusion and ionization of amino or carboxyl groups [33]. The data revealed that the solubility of soy protein increased from 34.15% to 37.35% by ACP treatment. Air, as the working gas during ACP treatment, is ionized and –OH and –NH can be generated. ACP treatment exposed the active sites on the surface of the soy protein and promoted the hydration reaction. Ji et al. [25] also found that ACP treatment could enhance solubility of peanut protein isolate. The swelling index of soy protein was increased after ACP treatment. Though some unstable compounds in soy protein film have been dissolved in the water after ACP treatment, swelling ability did not weaken as expected. The increase in swelling ability might be predominantly due to the improvement of hydrophilicity of soy protein film by the reactive species induced by ACP treatment, thus allowing more water to be absorbed. As for water content, it was observed that the ACP–treated soy protein film was 6.01% higher than that of the control group. Hydrogen bonds were more likely to be formed after ACP treatment, and thus, more water molecules could be locked in the film-forming process by the force of hydrogen bonds [5].

### 3.5. Thermal Properties of Soy Protein Film

Figure 6 and Table 2 presented the thermal profiles of the control and treated film by DSC. The denaturation temperature (Td) of the control sample exhibited was 113.4 °C. In contrast, the corresponding peak of the ACP–treated film shifted to 123.8 °C. There was a slight increase in Tg value, which increased from 98.2 °C to 100.3 °C after ACP treatment. It required a higher temperature and more heat for denaturation of ACP treatment film, indicating that the soy protein film was more stable after ACP treatment. The high–energy plasma discharge could break part of protein structure and generate some new bond [26]. ACP treatment would result in the increase in the hydrogen bonds of soy protein. During the discharge, the newly increased bonds of soy protein could enhance inter molecular force, leading to tighter cross-links between the proteins [35]. Jahromi et al. [36] reported that ACP can increase the thermal stability of casein membranes after treatment at 50 kV for 5 min.

### 3.6. Microstructure of Soy Protein Film

As shown in Figure 7, the surface of the control group was uneven and had obvious wrinkles under the scanning electron microscope. Besides, numerous cracks and pores could be detected for the control group. The uneven size of the protein clusters of the untreated sample, as well as the lack of good cross-linking between the molecules during film formation, resulted in the wrinkles, cracks, and pores. However, compared with the control counterpart, it could be observed the ACP-treated soy protein film has a smoother surface without visible cracks and pores. High–energy electrons excited by ACP decomposed the soy protein during ACP treatment, which changed its tertiary and quaternary structures. Changes in structure of soy protein had a momentous influence on the morphology of film. In general, ACP treatment would increase the roughness of the film surface observed by the scanning electron microscope. It has been stated by Chen et al. [37] that DBD plasma increased the roughness of zein film because of the intensification of the etching effect. The opposite phenomenon might be attributed to direct treatment on protein solution prior to film formation rather than on film itself. Soy protein would distribute more evenly in the solution, and the dispersed soy protein cross-linked into network by self-rearranging, which reduce the etching effect on the film. Dong et al. [38] evaluated zein in aqueous ethanol under ACP treatment, and they found plasma could decrease the aggregation degree of zein micelles. The observation of uniform surface with negligible pores and cracks in this finding could well explain for higher elongation at the break for the ACP–treated group. Furthermore, it was detected that elongation at the break remarkably increased with the ACP treatment in the above experiment. This might be related to the results of the scanning electron microscope, which indicated that ACP treatment on a protein solution largely lessened the cracks and pores. Moreover, the surface of the treated film became smoother. Hence, the observation of uniform surface with negligible pores and cracks in this finding could well explain for higher elongation at the break for the ACP–treated group.

### 3.7. Fourier Transform Infrared Spectroscopy Analysis of Soy Protein Film

The FT-IR results of the control and ACP–treated film are shown in Figure 8. An increase in the intensity of bands at 3200 cm^−1^ and 3400 cm^−^^1^ was observed after ACP treatment. The broad band caused by O–H stretching and N–H stretching was assigned to amide A of the soy protein film. The intensity of the C–H stretching band was observed at 2930 cm^−^^1^, and also increased. The carbonyl band appeared at 1668 cm^−^^1^, which was assigned to amide I. The amide II and amide III bands of soy protein were observed at 1546 cm^−^^1^ and 1256 cm^−^^1^, respectively [10]. The finding might be ascribed to the fact that considerable charged particles were generated by ACP, which bombard the surface of soy protein and bound to the active sites of soy protein [30]. Though no new peak was generated in both of the control and ACP treatment samples, the secondary structure components of two groups were slightly different. Hence, ACP treatment did not denature the soy protein or did not generate new functional groups [39]. Based on the results, it could be concluded that ACP treatment altered the film secondary structure involving the amide A region. The changes in the soy protein structure were caused by hydrogen bonds produced by high-energy particles, whereby hydrogen bonds helped to maintain the structure of the film. A similar phenomenon was also observed by Chen et al. [40], who found that the DBD-ACP pre-treatment led to a change in the protein conformation and promoted hydrogen bonding interactions between zein and polylactic acid.

### 3.8. Surface Roughness of Soy Protein Film

Surface morphology of the film was observed by an atomic force microscope (AFM). Maximum roughness (Rmax), root mean square roughness (Rms), and roughness average (Ra) were used to quantify the surface roughness changes [40]. As presented in Figure 9, the Rmax of the control and ACP-treated film were 162.7 nm and 88.4 nm, respectively. As summarized in Table 3, the Ra and Rms value of the control group were 17.1 nm and 22 nm, respectively. However, these two values appeared to be 10.9 nm and 13.4 nm, respectively. It was evident that the roughness of the film remarkably decreased after ACP treatment, which was inconsistent with others research. Pankaj et al. [26] demonstrated a dramatic increase in the surface roughness of the chitosan film subjected to DBD–ACP treatment. The main reason might be that film was directly exposed to ACP treatment in other research, while in this study, soy protein solution was disposed by ACP prior to preparing into the film. As is known, high–energy particles, including radicals, electrons, ions, neutrals, excited atoms, and UV radiations, exerted an etching effect on the surface of the film, which led to the increase in roughness. Nevertheless, direct treatment on the soy protein solution, rather than the film, avoided this etching impact, which in turn effectively reduced the damage to the film and thus resulted in the reduction of the roughness in our present investigation. Dong et al. [38] demonstrated that the discharge space, the shockwave, as well as high-energy particles induced by ACP treatment, might cause a breakage in intermolecular forces between zein micelles, and thus disperse the originally clumped zein protein in aqueous ethanol. Similarly, the soy protein solution treated by ACP was more uniformly dispersed in the solution after ACP treatment. The direct treatment on the protein was more favorable for its rearrangement to form a regular and uniform film structure. These results were well in accordance with those obtained from scanning electron microscope analysis of the film.

### 3.9. Surface Hydrophilicity of Soy Protein Film

The water contact angle (WCA), an important variable, was used to evaluate the hydrophilicity and hydrophobicity of the film surface. A higher WCA stands for stronger hydrophobicity of the film [4]. As is known, soy protein film exhibited strong hydrophobicity [5]. Low WCA value of the film was observed in the case of ACP treatment, while relatively higher value of WCA was obtained for the control sample (Figure 10). Based on Table 4, the WCA of the control and ACP–treated film was 87.9°and 77.2°, respectively. The deduction was presumably owing to the interaction of the active particles with the soy protein, leading to the increase in polarity [29]. Based on FT–IR analysis, a large number of –OH groups and hydrogen bonds were formed on the film after ACP modification. The hydrophilic groups on the protein surface would result in hydration by the water molecules [41]. A number of studies indicated enhancement of surface roughness occurred with the improved surface hydrophilicity when the film was directly subjected to ACP treatment [42]. As afore mentioned, direct treatment on film itself might lead to a certain damage to the barrier properties. In terms of AFM observation in our study, surface roughness of the film decreased due to the pretreatment on the soy protein solution rather than the film. Furthermore, increased values of solubility, swelling index, and moisture content illustrated the stronger hydrophobicity, which was inconsistent with the lower WCA, which also indicated the better affinity for water molecules. A low water contact angle and high hydrophilic character of film were obtained for ACP treatment [43]. Therefore, the increase of hydrophilic groups could account for the decrease in WCA. The proposed method of using ACP to modify the soy protein could effectively improve the surface hydrophilicity of the film.

## 4. Conclusions

Though ACP has been successfully applied to improve the packaging properties of protein film, few data can be achieved concerning direct modification of a protein solution by plasma. In this investigation, a soy protein solution was subjected to DBD-ACP treatment prior to preparation in the edible film. The optimal conditions were found for soy protein (10%), glycerol (2.8%), and ACP treatment at 30 kV for 3 min, under which, changes in physicochemical and structure characteristics were observed. An enhanced elongation at thebreak and reduced WVP occurred, accompanied by the reinforcement of thermal properties, demonstrating the cross-linking with the soy protein matrix. ACP pretreatment also resulted in the increase of hydrophilicity, which was proved by the improvement of So, Si, MC, and the decline of the water contact angle. Moreover, compared with the control group, ACP treatment could obviously reduce the voids and cracks of the film by SEM images. Fourier transform infrared spectroscopy suggested that ACP pretreatment would form more hydrophilic groups, such as hydroxyl groups and hydrogen bonds. Interestingly, roughness decreased for the ACP treated sample, owing to the direction treatment on the protein solution rather than the film. Therefore, it was expected that ACP treatment on the protein solution could serve as a feasible and effective surface modification approach that would enlarge edible film application in the package industry with desirable functions.

## Figures and Tables

**Figure 1 foods-11-01196-f001:**
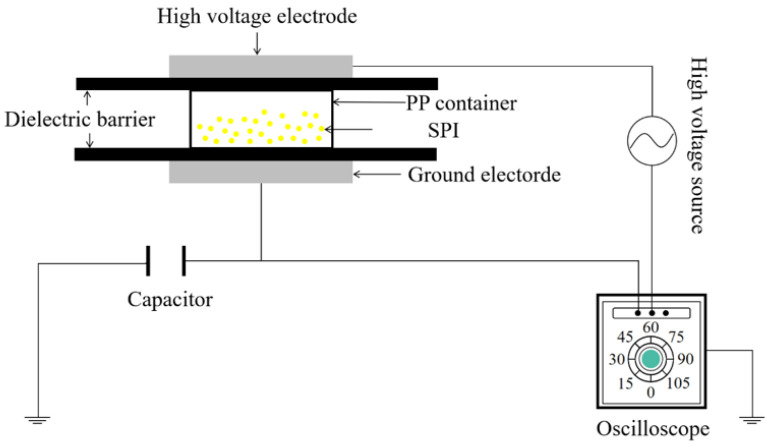
Schematic diagram of experimental equipment for DBD plasma system.

**Figure 2 foods-11-01196-f002:**
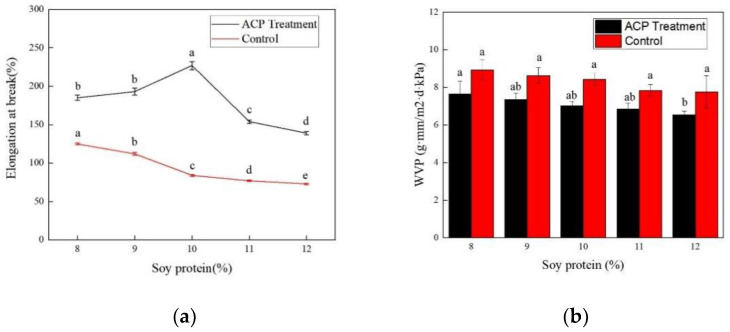
Effect of soy protein concentration on mechanical properties of the film. (**a**) Elongation at break of ACP–treated and control group. (**b**) WVP of ACP–treated and control group. Different lowercase letters denote significant differences (*p* < 0.05).

**Figure 3 foods-11-01196-f003:**
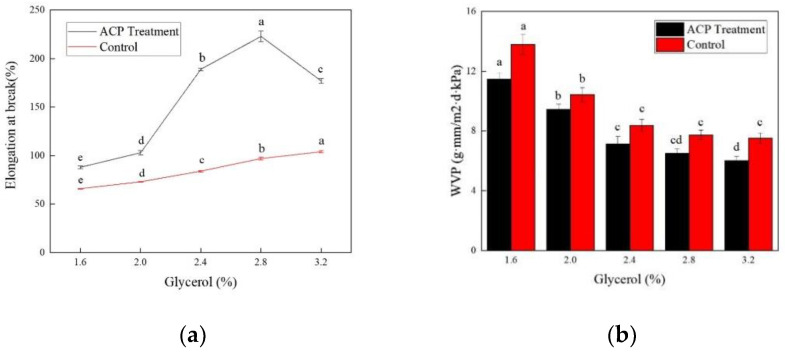
Effect of glycerol concentration on elongation at break and WVP of the soy protein film. (**a**) Elongation at break of ACP–treated and control group. (**b**) WVP of ACP–treated and control group. Different lowercase letters denote significant differences (*p* < 0.05).

**Figure 4 foods-11-01196-f004:**
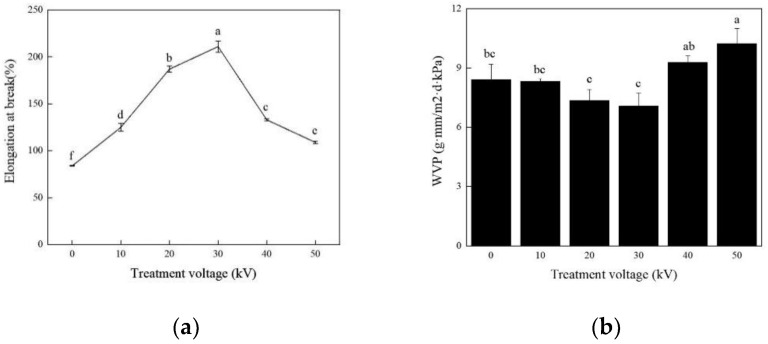
Effects of ACP treatment voltages on elongation at break and WVP of the soy protein film. (**a**) Elongation at break; (**b**) WVP. Different lowercase letters denote significant differences (*p* < 0.05).

**Figure 5 foods-11-01196-f005:**
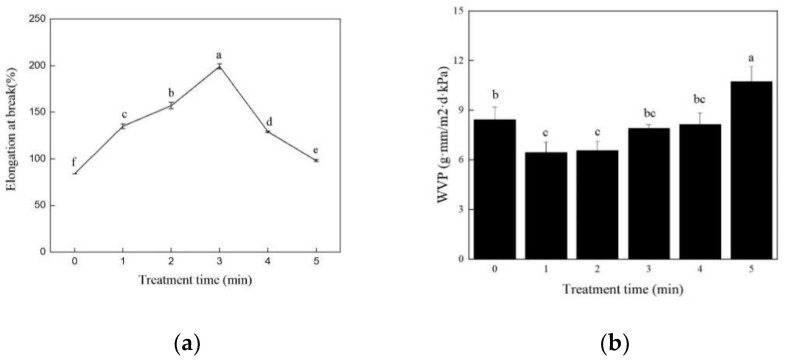
Effects of different ACP treatment times on elongation at break and WVP of the soy protein film. (**a**) Elongation at break; (**b**) WVP. Different lowercase letters denote significant differences (*p* < 0.05).

**Figure 6 foods-11-01196-f006:**
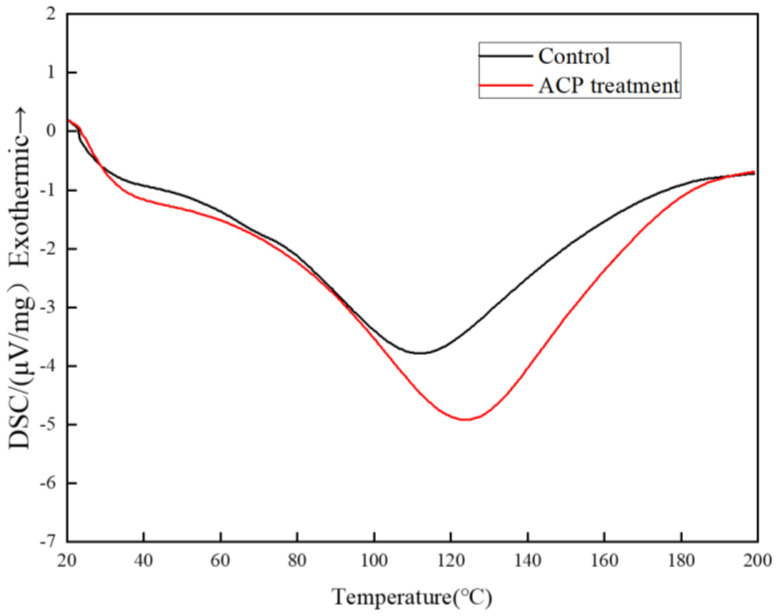
DSC curve of control and ACP–treated film at 30 kV for 3 min. Untreated soy protein film was considered as the control. The amount of soy protein was 10% and glycerol was 2.8%.

**Figure 7 foods-11-01196-f007:**
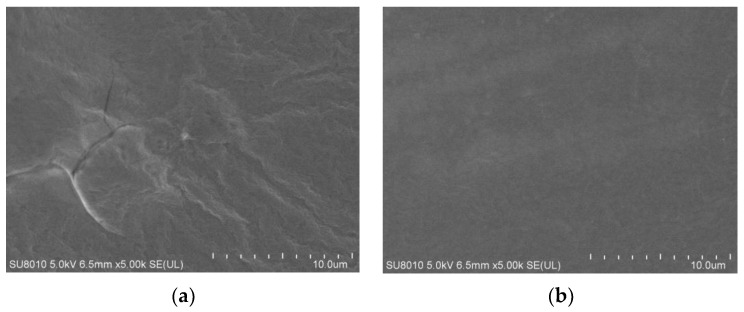
Scanning electron microscope images of the ACP–treated film and control. (**a**) Untreated soy protein film. (**b**) Soy protein film treated by ACP at 30 kV for 3 min. Untreated soy protein film was considered as the control. The amount of soy protein was 10% and glycerol was 2.8%.

**Figure 8 foods-11-01196-f008:**
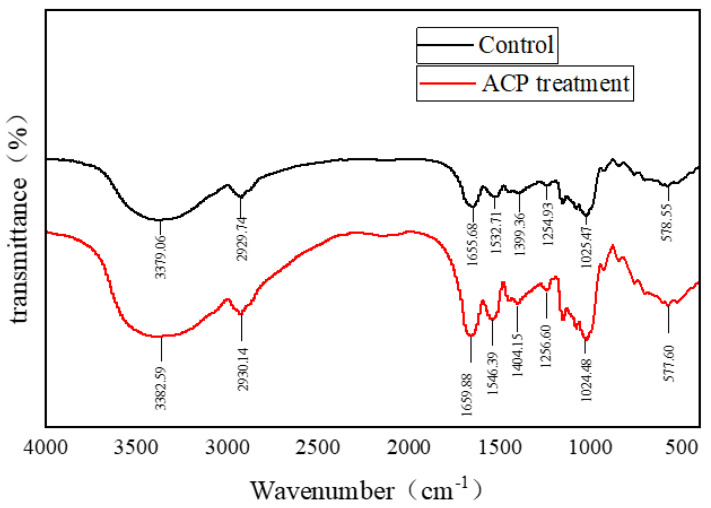
Fourier near-infrared spectra of control and ACP–treated film at 30 kV for 3 min. Untreated soy protein film was considered as the control. The amount of soy protein was 10% and glycerol was 2.8%.

**Figure 9 foods-11-01196-f009:**
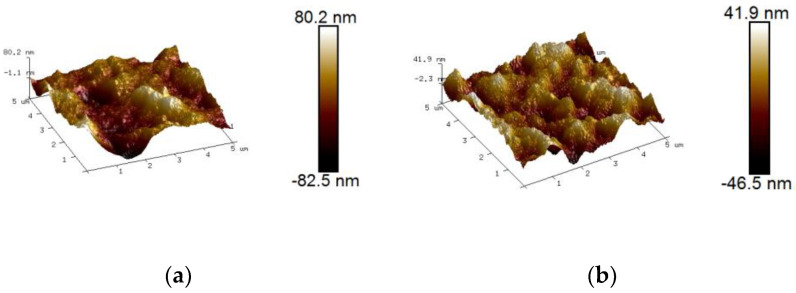
Atomic force microscope images of control and ACP–treated film at 30 kV for 3 min. (**a**) Untreated soy protein film. (**b**) Soy protein film treated by ACP at 30 kV for 3 min. Untreated soy protein film was considered as the control. The amount of soy protein was 10% and glycerol was 2.8%.

**Figure 10 foods-11-01196-f010:**
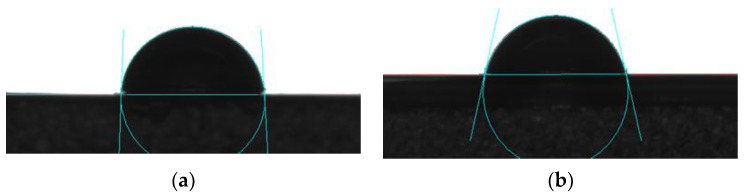
Photos of water contact angle of the control and ACP–treated samples. (**a**) Untreated soy protein film. (**b**) Soy protein film treated at 30 kV for 3 min. Untreated soy protein film was considered as the control. The amount of soy protein was 10% and glycerol was 2.8%.

**Table 1 foods-11-01196-t001:** Solubility, swelling index, and moisture content of soy protein film. ACP treatment is soy protein film treated for 3 min, at 30 kV. The control is untreated soy protein film.

Stage	So (%)	Si (%)	MC (%)
Control	34.15 ± 0.07	134.89 ± 0.41	25.13 ± 0.16
ACP treatment	37.35 ± 0.15	153.52 ± 1.32	31.44 ± 0.21

So: Solubility. Si: swelling index. MC: moisture content. The values are expressed as mean ± SD. (*n* = 3).

**Table 2 foods-11-01196-t002:** Thermodynamic parameters of control and ACP–treated film, with the amount of soy protein being 10% and glycerol being 2.8%.

Thermodynamic Parameters	Tg (°C)	Td (°C)	Peak Area (μVs/mg)	Peak (μV/mg)	Enthalpy (J/g)
Control	98.2	113.4	512.6	3.7829	10.338
ACP treatment	100.3	123.8	674.9	4.9177	14.025

**Table 3 foods-11-01196-t003:** Roughness parameters for the control and ACP–treated samples with the amount of soy protein being 10% and glycerol being 2.8%.

Roughness Index	Rmax (nm)	Ra (nm)	Rms (nm)
Control	162.7 ± 2.14	17.1 ± 0.45	22.0 ± 0.54
ACP treatment	88.4 ± 1.33	10.9 ± 27	13.4 ± 0.19

The values are expressed as mean ± SD. (*n* = 3).

**Table 4 foods-11-01196-t004:** Water contact angle of the control and ACP–treated samples. The ACP treatment is soy protein film treated at 3 min, at 30 kV. The control is untreated soy protein film.

Water Contact Angle	Right	Left
Control	86.9° ± 0.53	87.9° ± 0.47
ACP treatment	78.7° ± 0.32	77.2° ± 0.39

The values are expressed as mean ± SD. (*n* = 3).

## Data Availability

The data that support the findings of this study are available from the corresponding author at reasonable request.

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
