# Peer review of "Surface Modification via Dielectric Barrier Discharge Atmospheric Cold Plasma (DBD–ACP): Improved Functional Properties of Soy Protein Film"

_foods, 2022, doi:10.3390/foods11091196_

Round 1

Reviewer 1 Report

The overall writings are good. 

However, a detailed experimental design regarding the different aspects tested (independent variables) and also the selection criteria for optimisation (to best parameters) before proceeding to the next stage should be made clear. 

For example, from the different voltages, how do you select the best to be used as the fixed voltage in the next experiment. 

For result and discussions, part 3.1 and 3.2, since there are 4 aspects to be discussed, it is suggested to segregate the discussion for a, b, c and d, with each detailed discussion on the effect/trend and also supporting references, respectively; instead of combining all in 1-2 long paragraphs. 

Figure 5- to remove the word refers to for (a) and (b).

Section 3.6, detailed explanation to be added to support figure 7 and Table 2, respectively. Same thing applies to 3.7. 

In terms of 

Author Response

Point-by-point response to the comments of reviewer 1

Dear reviewer, we are pleased to receive your comments and suggestions, and sincerely thank you for these. We have revised our manuscript according to your comments as we explain below in our point-by-point responses.

  1. For result and discussions, part 3.1 and 3.2, since there are 4 aspects to be discussed, it is suggested to segregate the discussion for a, b, c and d, with each detailed discussion on the effect/trend and also supporting references, respectively; instead of combining all in 1-2 long paragraphs.

Response: 1 Thank you for your suggestion. It sounds really reasonable since part 3.1 and 3.2 seemed a little bit confused in the original MS. As suggested, the 4 aspects, namely the 4 factors, including soybean protein concentration, glycerol concentration, ACP treatment voltage and exposure time, were discussed separately in terms of elongation at break and water vapor permeability. The 4 factors appeared as 4 detailed discussions on the effect and also supporting references in the revised MS instead of combining all in 1-2 long paragraphs. The current one was more clearly and easy to understand. The title of this part was “3.1 Factors affecting elongation at break and water vapor permeability of the soy protein film” and the subtitles were “3.1.1. Effect of soybean protein concentration, 3.1.2. Effect of glycerol concentration, 3.1.3 Effects of ACP treatment voltage, and 3.1.4. Effects of ACP treatment time. (Page 5-8 lines 174-284).

  1. Figure 5- to remove the word refers to for (a) and (b).

Response: 2     Done accordingly. The words “refers to” were deleted in all the figure captions.

  1. Section 3.6, detailed explanation to be added to support figure 7 and Table 2, respectively. Same thing applies to 3.7.

Response: 3    Indeed. The explanation was too simple in the original MS. We appreciate your comment. We read the relevant literature and detailed explanations were supplemented to support Figure 7 and Table 2, respectively. Meanwhile, we also discussed Section 3.7 in detail. The relevant passage from the revised manuscript reads as follows:

"The main reason might be that film was directly exposed to ACP treatment in other researches, while in this study, soy protein solution was disposed by ACP prior to preparing into the film. As is known, high-energy particles, including radicals, electrons, ions, neutrals and excited atoms and UV radiations, exerted etching effect on the surface of the film, which led to the increase in roughness. Nevertheless, direct treatment on the soy protein solution, rather than the film, avoided this etching impact, which in turn effectively reduced the damage to the film and thus resulted in the reduction of the roughness in our present investigation. Dong et al. [34] demonstrated that the discharge space, the shockwave, as well as high energy particles induced by ACP treatment, might cause a breakage in intermolecular forces between zein micelles and thus disperse the originally clumped zein protein in aqueous ethanol. Similarly, soy protein solution treated by ACP was more uniformly dispersed in the solution after ACP treatment. The direct treatment on the protein was more favorable for its rearrangement to form a regular and uniform film structure. These results were in well accordance with those obtained from scanning electron microscope analysis of the film." (Page 11 lines 362-377).

"Table 2. Roughness parameters for the control and ACP treated samples. ACP treatment is soy protein film treat at 3 min, 30 kV. Control is untreated soy protein film. " (Page 12 lines 381-382).

" The deduction was presumably owing to the interaction of the active particles with the soy protein, leading to the increase in polarity [29]. Based on FT-IR analysis, a large number of -OH groups and hydrogen bonding were formed on the film after ACP modification. The hydrophilic groups on the protein surface would result in hydration by the water molecules [38]. " (Page 12 lines 390-394).

" Furthermore, increased values of solubility, swelling index and moisture content illustrated the stronger hydrophobicity, which was inconsistent with the lower WCA, which also indicated the better affinity for water molecules. Lowe water contact angle and high hydrophilic character of film were obtained for ACP treatment [40]. "(Page 12 lines 399-402).

" Table 3. Water contact angle of the control and ACP treated samples. ACP treatment is soy protein film treat at 3 min, 30 kV. Control is untreated soy protein film. "(Page 13lines 409-410).

Reviewer 2 Report

I reviewed the article which entitled: Surface Modification via Dielectric Barrier Discharge Atmospheric Cold Plasma (DBD-ACP): Improved Functional Properties of Soy Protein Film. There is a huge concern regarding the novelty of the presented work, the author neither justified the novelty nor their findings.

The abstract needs to be improved to be a sharp and standalone. The current did not represent the observed values, please keep reminded that abstract should represent all sections of manuscript.

Solubility, swelling and moisture content should be added to the manuscript.

Figure quality is very low in some figures.

Figure 3, significant letter should be added on the columns.

Standard deviation and significant letter should be added in the tables.

Discussion of different parts seems introduction, lacks relation between feedings of studies and previous results; conclusions are not drawn appropriately.

Author Response

Point-by-point response to the comments of reviewer 2

Dear reviewer, we are pleased to receive your comments and suggestions, and sincerely thank you for these. Thank you for your good advice to help us improve the quality of the paper. We have revised our manuscript according to your comments as we explain below in our point-by-point responses.

The abstract needs to be improved to be a sharp and standalone. The current did not represent the observed values, please keep reminded that abstract should represent all sections of manuscript.

Response: 1 we thank you very much for giving us an opportunity to revise our manuscript. Thank you for your waiting. Thank you for your recognition of our work and give us the chance to improve our manuscript. We were sorry for the inappropriate description. We appreciate this comment and have added a correct of your suggestion follows:

" Atmospheric cold plasma (ACP), a novel technology, has been widely adopted as an efficient approach in surface modification of the film. The effect of ACP treatment on the physicochemical and structural properties of soy protein film were investigated. As a result, the optimal conditions for the preparation of the film were determined to soy protein (10%), glycerol (2.8%), ACP treatment at 30 kV for 3min on the basis of elongation at break and water vapor permeability. Under the optimal conditions, the ACP treated films exhibited enhanced polarity according to the increased values of solubility, swelling index and moisture content, compared with the untreated counterpart. Increase in the hydrophilicity is also confirmed by the water contact angle analysis, which decreased from 87.9° to 77.2° after ACP pretreatment. Thermostability was also improved by ACP exposure in terms of DSC analysis. SEM images confirmed the tiny pores and cracks on the surface of film could be lessened by ACP pretreatment.  Variations in the Fourier transform infrared spectroscopy indicated that some hydrophilic groups were formed by ACP pretreatment. Atomic force microscopy data revealed that the roughness of soy protein film which was pre-treated by ACP was lower than that of control group, with a Rmax value of 88.4 nm and 162.7 nm for the ACP treated and untreated samples, respectively. The soy protein film has been charac-terized structurally by FT-IR and DSC as well as morphological characterization was done by SEM and AFM. The soy protein film by ACP modified was more stable than the control group. Hence, the great potential in improving the properties of the film enables ACP treatment a feasible and promising alternative to other modification methods. "(Page 1 lines 9-27).

  1.   Solubility, swelling and moisture content should be added to the manuscript.

Response: 2    Thank you for this important question. Thank you for your good advice to help us improve the quality of the paper. we completed the addition and analysis of three important indexes: solubility, swelling index and moisture content. This is described in the text as follows:  

""Soy protein films were weighed before and after drying in an electric ovenat 110 ℃ until a constant weight was obtained. MC was expressed as Equation (3). "(Page 4 lines 142-143).

"Soy protein film (2 cm × 2 cm) was soaked in 30 mL of pure water. Then, the sample was placed in an oscillation box at 25 ℃for 24 h and dried to a constant weight. The solubility was calculated as Equation (4): "(Page 4 lines 147-149).

"Film pieces (2 cm × 2 cm) were dried at 70 ℃ for 24 h in a vacuum oven to obtain the initial dry weight. The sample was transferred into a 100 ml beaker with30 ml pure water and soaked in the water at 25 ℃ for 24 h Subsequently, the moisture on the film surface was sucked with filter paper and weighed up. The swelling index was expressed as Equation (5)."(Page 4 lines 153-157).

"Moisture content, solubility and swelling index are three important characteristics of protein film, which can positively reflect the hydrophilicity of soy protein film. The results of solubility, swelling index and water content of soy protein film are shown in Table 1. By comparing ACP treatment film and control, ACP treatment had a greater degree effect on the solubility and swelling than untreated. The solubility was related to the hydrophilic properties of films and the swelling index of film involved water dif-fusion, ionization of amino or carboxyl groups [34]. The data revealed that the solubility of soy protein has increased (from 34.15% to 37.35%) by ACP treatment (3min,30 kV). During ACP treatment, air is ionized to produce -OH and - NH. ACP treatment ex-posed the active sites on the surface of soy protein and promoted the hydration reac-tion. Ji et al. [25] found that ACP treatment could promoted solubility of peanut pro-tein isolate. The swelling index of soy protein was increased after ACP treatment. Be-cause ACP treatment improves the hydrophilicity of soy protein film, some unstable structures of soy protein film have been dissolved in water, which still could not weaken the swelling ability of ACP treatment film. By measuring the water content of soybean protein film, it was found that the ACP treated soy protein film was 6.01% higher than that of the control group. Soy protein is more likely to form hydrogen bonds after ACP treated, and more water molecules could be locked in the film-forming engineering through the force of hydrogen bonds [5]. "(Page 8 lines 269-288).

" Table 1. Solubility, swelling index and moisture content of soy protein film. ACP treatment is soy protein film treat at 3 min ,30 kV. Control is untreated of soy protein film."(Page 8 lines 289-290).

  1. Figure quality is very low in some figures.

Response: 3    Indeed. Done accordingly. Thank you for your advice. We added lowercase letters representing different significance in Figure 2 and Figure 3.

  Figure 3, significant letter should be added on the columns.

Response: 4    Thank you for your advice. Thank you for your good advice to help us improve the quality of the paper. We added lowercase letters representing different significance in Figure 2 and Figure 2.

  Standard deviation and significant letter should be added in the tables.

Response:5     We were sorry for our inaccurate expression. We appreciate the reviewer’s suggestion. According to the reviewer’s comment (Comment 2), we have adjusted the order of relevant tables. The standard deviation calculated from three parallel experiments were supplemented in the revised MS. Meanwhile, the significant letter representing different significance should also be added. However, it was unable to obtain the significant letter since there were only two groups. Please check our revised manuscript.

Revised content: table 1, table 3 and table 4.

  Discussion of different parts seems introduction, lacks relation between feedings of studies and previous results; conclusions are not drawn appropriately.

Response:6    Actually, the discussion did seem introduction. Especially, it lacked the relation between findings of studies and previous results. Thanks for the reviewer’s advice. We were sorry for the inappropriate expression. We had revised the discussion part and paid more attention to the relation between our study and previous studies, and carefully compared our findings with other researches. We also analyzed the inner relationship between the results we obtained (Page 13 lines 416-433). Meanwhile, as suggested by the reviewer’s advice, the inappropriate conclusions were redrawn in the revised MS as follows:

Revised sentence: "Though ACP has been successfully applied to improve the packaging properties of protein film, few data can be achieved concerning direct modification of protein solution by plasma. In this investigation, soy protein solution was subjected to DBD-ACP treatment prior to preparation into the edible film. The optimal conditions were found to soy protein (10%), glycerol (2.8%), ACP treatment at 30 kV for 3min, under which, changes in physicochemical and structure characteristics were observed. An enhanced elongation at break and reduced WVP occurred, accompanied by the reinforcement of thermal properties, demonstrating the cross-linking with soy protein matrix. ACP pretreatment also resulted in the increase of hydrophilicity, which was proved by the improvement of So, Si, MC and the decline of the water contact angle. Moreover, compared with the control group, ACP treatment could obviously reduce the voids and cracks of the film by SEM images. Fourier transform infrared spectroscopy suggested that ACP pretreatment would form more hydrophilic groups, such as hydroxyl groups and hydrogen bonds. Interestingly, roughness decreased for the ACP treated sample, owing to the direction treatment on the protein solution rather than the film. Therefore, it was expected that ACP treatment on protein solution could serve as a feasible and effective surface modification approach that would enlarge edible film application in package industry with desirable functions. " (Page 13 lines 416-433).

Reviewer 3 Report

Please specify the amount of soy protein and glycerol used for the results in Figure 2b, 2c, 3b, 3c, 4 to 8, and Table 2 and 3. 

Your electron microscopy result showed that the plasma treatment reduced the pores and defects on the protein film. Do you think the improvement in the microstructure has an effect on your elongation at break results? If you can detect any relation between the microstructure and the elongation at break results, could you please discuss their relationship in the paper?

 There are a few spelling errors in the manuscript. Please check the spelling. 

Author Response

Point-by-point response to the comments of reviewer 3

Dear reviewer, we are pleased to receive your comments and suggestions, and sincerely thank you for these. We have revised our manuscript according to your comments as we explain below in our point-by-point responses.

  1. Please specify the amount of soy protein and glycerol used for the results in Figure 2b, 2c, 3b, 3c, 4 to 8, and Table 2 and 3.

Response: 1Thank you very much for giving us an opportunity to revise our manuscript. Thank you for your waiting. Indeed, it was not clear enough without the amount of soy protein and glycerol. As suggested, the amount of soy protein and glycerol used for the results in Figure 2b, 2c, 3b, 3c, 4 to 8, and Table 2 and 3 was specified as follows:

" Figure 2. Effect of different preparation conditions on elongation at break of soy protein film. (a) effect of soy protein concentration with the addition of glycerol of 2.4 %, ACP treatment time of 3 min and treatment voltage of 30 kV (b) effect of ACP treatment voltage with the addition of soy protein of 10 %, glycerol of 2.4% and treatment time for 3 min. (c) effect of ACP treatment time with the addition of soy protein of 10 %, glycerol of 2.4% and treatment voltage of 30 kV. (d) effect of glycerol concentration with the addition of soy protein of 10 %, ACP treatment time of 3 min and treatment voltage of 30 kV. "(Page 5 lines 179-185).

"Figure 3. Effect of different preparation conditions on water vapor transmission rate of soy protein film. (a) effect of soy protein concentration with the addition of glycerol of 2.4 %, ACP treatment time of 3 min and treatment voltage of 30 kV (b) effect of ACP treatment voltage with the addition of soy protein of 10 %, glycerol of 2.4% and treatment time for 3 min. (c) effect of ACP treatment time with the addition of soy protein of 10 %, glycerol of 2.4% and treatment voltage of 30 kV. (d) effect of glycerol concentration with the addition of soy protein 10 %, ACP treatment time of 3 min and treatment voltage of 30 kV."(Page 7 lines 242-248).

"Figure 4. DSC curve of control and ACP treated film at 30 kV for 3 min. Untreated soy protein film was considered as control. The amount of soy protein was 10 % and glycerol was 2.8%."(Page 8 lines 268-269).

" Table 1. Thermodynamic parameters of control and ACP treated film with the amount of soy protein 10% and glycerol 2.8%."(Page 8 lines 270).

"Figure 5.  Scanning electron microscope images of the ACP treated film and control. (a)refers to untreated soy protein film. (b)refers to soy protein film treated by ACP at 30 kV for3 min. Untreated soy protein film was considered as control. The amount of soy protein was 10 % and glycerol was 2.8%."(Page 9 lines 296-298).

"Figure 6.  Fourier near-infrared spectra of control and ACP treated film at 30 kV for 3 min. Untreated soy protein film was considered as control. The amount of soy protein was 10 % and glycerol was 2.8%."(Page 10 lines 317-319).

"Figure 7. Atomic force microscope images of control and ACP treated film at 30 kV for 3 min. (a)refers to untreated soy protein film. (b)refers to soy protein film treated by ACP at 30 kV for 3 min. Untreated soy protein film was considered as control. The amount of soy protein was 10 % and glycerol was 2.8%."(Page 11 lines 345-347).

" Table 2. Table 2. Roughness parameters for the control and ACP treated samples with the amount of soy protein 10 % and glycerol 2.8%."(Page11 lines 349-350).

"Figure 8. Figure 8. Photos of water contact angle of the control and ACP treated samples. (a)refers to untreated soy protein film. (b)refers to soy protein film treated at 30 kV for 3 min. Untreated soy protein film was considered as control. The amount of soy protein was 10 % and glycerol was 2.8%."(Page 12 lines 368-370).

"Table 3. Water contact angle of the control and ACP treated samples with the amount of soy protein 10 % and glycerol 2.8%."(Page 12 lines 371-372).

  1. Your electron microscopy result showed that the plasma treatment reduced the pores and defects on the protein film. Do you think the improvement in the microstructure has an effect on your elongation at break results? If you can detect any relation between the microstructure and the elongation at break results, could you please discuss their relationship in the paper?

Response: 2    Really a good question. We appreciate the reviewer suggestion. Actually, plasma treatment reduced the pores and defects on the protein film by SEM results. We checked the relation between microstructure and elongation at break and found that there was a correlation between them. However, it was not explicitly discussed in the original MS and the relationship was discussed in the revised MS as follows:

"Furthermore, it was detected that elongation at break remarkably increased with the ACP treatment in the above experiment. This might be related to the results of the scanning electron microscope, which indicated that ACP treatment on protein solution largely lessened the cracks and pores. Moreover, the surface of the treated film became smoother. Hence, the observation of uniform surface with negligible pores and cracks in this finding could well explain for higher elongation at break for ACP treated group." (Page 9 lines 289-395).

  1. There are a few spelling errors in the manuscript. Please check the spelling.

Response: 4    Indeed. Thank you for your advice. There were a few spelling errors in the original MS due to carelessness. We have double read the MS and carefully checked the usage of language, as well as the spelling.

Round 2

Reviewer 1 Report

  1. Line 64 and 65- scientific name italic
  2. Line 72. Sentence with only 1 word?
  3. Line 74, spacing
  4. Line 137 Moisture; space after oven
  5. Line 149 space after with
  6. Reference required for 2.2, 2.3, 2.5, 2.6, 2.7, 2.8, 2.9, 2.10, 2.11, 2.12
  7. Line 171- check cm-1 superscript
  8. Line 178- check m2
  9. Line 231 font
  10. 3.1.3 should be 3.2?
  11. 3.1.4 should be 3.3?
  12. 3.3 should be 3.4 and same for subsequent changes?
  13. Line 330-331- check degree celcius sign
  14. Line 397- Figure 8
  15. Figure 7, 9& 10- remove the word refers to

Author Response

Point-by-point response to the comments of reviewer 1

Dear reviewer,

Thank you for your suggestion. we are pleased to receive your comments and suggestions, and sincerely appreciate for these. We thank you for your hard work. We apologize for the format and references. We revised the problems existing in the manuscript. The specific amendments are as follows:

Comment 1: Line 64 and 65- scientific name italic

Response 1:   Thank you for your suggestion. Indeed, the names of the bacteria should be italic and they were revised as follows: Escherichia coli [17], Listeria [18], Salmonella, Shigella [19], Staphylococcus aureus [20], Thermobacteria [21].

(Page 5 lines 65-66).

Comment 2:  Line 72. Sentence with only 1 word?

Response 2 :  We appreciate the reviewer’s carefulness. This is an error due to our carelessness and the word “Enhancing” and the stop “.” was deleted in our revised MS.  (Page 2 lines 72).

Comment 3:  Line 74, spacing

Response 3:   Done accordingly. There should be space between " enhanced” and  "for ". (Page 2 lines 74).

Comment 4:  Line 137 Moisture; space after oven

Response 4:  Done accordingly. The space was added between the word “oven” and “at” . (Page 4 lines 137-138).

Comment 5: Line 149 space after with

Response 5:  Indeed. The space was added after the word “with”. ( Page 4 lines 152).

Comment 6:  Reference required for 2.2, 2.3, 2.5, 2.6, 2.7, 2.8, 2.9, 2.10, 2.11, 2.12

Response 6:  Indeed. Thank you very much for your suggestion. Corresponding references were supplemented for those methods in the revised MS.

Comment 7:  Line 171- check cm-1 superscript

Response 7:  Done accordingly. “cm-1” was changed to “cm-1” in the revised MS. ( Page 5 lines 175).

Comment 8: Line 178- check m2

Response 8:  Indeed. We apologize for our careless mistake. The correct format should be μm2. ( Page 5 lines 183).

Comment 9: Line 231 font

Response 9:  Done accordingly. The font of “with glycerol addition of 2.8%” was revised. ( Page6 lines 236).

Comment 10:  3.1.3 should be 3.2? 3.1.4 should be 3.3? 3.3 should be 3.4 and same for subsequent changes?

Response 10: Thank you for your suggestions. Your comment seems rather reasonable. The order of 3.1.3 and 3.1.4 were changed to 3.2 and 3.3, respectively. Meanwhile, all the orders of subsequent subheading were adjusted in the revised MS.

Comment 11: Line 330-331- check degree celcius sign

Response 11:  Indeed. We checked degree celcius sign and all of them were changed to “°C” in the revised MS. ( Page9 lines 337-338).

Comment 12:  Line 397- Figure 8

Response 12:  Done accordingly. The font of the caption in Figure 8 was united in the revised MS. ( Page12 lines 401-403).

Comment 13: Figure 7, 9& 10- remove the word refers to

Response 13:  Thank you for your suggestion. The words “refers to” in Figure 7, 9& 10 were deleted in the revised MS.

Reviewer 2 Report

accept

Author Response

Dear reviewer

Thank you for your suggestions and comments, and sincerely thank you for these, which has greatly improved the quality of the manuscript. We appreciate your approval of our revision work. We thank you for your hard work. 

On behalf of my co-authors, we thank you and the reviewers very much for giving us an opportunity to revise our manuscript. We also appreciate you and reviewers very much for their positive and constructive comments, and suggestions on our manuscript (foods-1632134).

We sincerely looking forward to your reply, thank you.

Yours sincerely,

Jing Chen

Corresponding author: Jing Chen,
